# Applying UHPLC-HRAM MS/MS Method to Assess Host Cell Protein Clearance during the Purification Process Development of Therapeutic mAbs

**DOI:** 10.3390/ijms25179687

**Published:** 2024-09-07

**Authors:** Reiko Kiyonami, Rafael Melani, Ying Chen, AI De Leon, Min Du

**Affiliations:** 1Thermo Fisher Scientific, Lexington, MA 02421, USA; min.du@thermofisher.com; 2Thermo Fisher Scientific, San Jose, CA 95134, USA; rafael.melani@thermofisher.com; 3Thermo Fisher Scientific, Bedford, MA 01730, USA; ying.chen@thermofisher.com (Y.C.); al.deleon@thermofisher.com (A.D.L.)

**Keywords:** host cell proteins, HPLC MS/MS method, process related HCP profiling

## Abstract

Host cell proteins (HCPs) are one of the process-related impurities that need to be well characterized and controlled throughout biomanufacturing processes to assure the quality, safety, and efficacy of monoclonal antibodies (mAbs) and other protein-based biopharmaceuticals. Although ELISA remains the gold standard method for quantification of total HCPs, it lacks the specificity and coverage to identify and quantify individual HCPs. As a complementary method to ELISA, the LC-MS/MS method has emerged as a powerful tool to identify and profile individual HCPs during the downstream purification process. In this study, we developed a sensitive, robust, and reproducible analytical flow ultra-high-pressure LC (UHPLC)-high-resolution accurate mass (HRAM) data-dependent MS/MS method for HCP identification and monitoring using an Orbitrap Ascend BioPharma Tribrid mass spectrometer. As a case study, the developed method was applied to an in-house trastuzumab product to assess HCP clearance efficiency of the newly introduced POROS™ Caprylate Mixed-Mode Cation Exchange Chromatography resin (POROS Caprylate mixed-mode resin) by monitoring individual HCP changes between the trastuzumab sample collected from the Protein A pool (purified by Protein A chromatography) and polish pool (purified by Protein A first and then further purified by POROS Caprylate mixed-mode resin). The new method successfully identified the total number of individual HCPs in both samples and quantified the abundance changes in the remaining HCPs in the polish purification sample.

## 1. Introduction

Host cell proteins (HCPs) are process-related impurities produced by the host organism during biotherapeutic manufacturing and production. HCP contaminants in a biotherapeutic drug product can cause issues with immunogenicity, off-target biological effects, or enzymatic activity such as degradation of the active pharmaceutical ingredient or formulation excipients. Due to product safety and efficacy risks, a drug product’s overall quantity of residual HCPs as well as the presence of high-risk HCPs is a critical quality attribute (CQA) that must be characterized and controlled throughout the manufacturing process [1,2]. Enzyme-linked immunosorbent assay (ELISA) is often used for HCP detection tests, as it can provide high throughput and high sensitivity for quantification of total HCPs. However, it lacks the specificity and coverage to identify and quantify individual HCPs. The LC-MS/MS method has emerged as an orthogonal method for HCP analysis to identify and quantify individual HCPs in a process step or even the final product [3,4,5]. Knowledge about the identities and their relative abundances of individual HCPs can be used to identify critical HCPs (such as high-risk HCPs and difficult-to-remove HCPs), providing useful information for HCP risk assessment to assure the safety and efficacy of therapeutic drug products [6,7,8]. In addition, the ability to rapidly measure HCPs and follow their clearance throughout the downstream process can be used to pinpoint sources of HCP contamination, helping to optimize biopharmaceutical production processes to minimize HCP levels [9,10,11].

The key challenge to applying the LC-MS/MS method for HCP analysis is how to address the huge intrasample dynamic range of pharmaceutical products. The concentrations of HCPs are significantly lower than the targeted therapeutic proteins, making it very challenging to detect low-abundance (<10 ppm) HCPs among the dominant therapeutic proteins. To overcome this challenge, efforts have been made to optimize sample preparation to improve the dynamic range, such as increasing HCP abundances relative to the abundance of the drug substance [12,13,14,15,16], removal of mAbs by affinity depletion [17,18], enhancing separation of protein digests by 2D-online or offline fractionation [19,20,21,22], and separating HCPs from the mAbs by molecular weight cut-off [23,24]. Although these strategies demonstrated successful detection of single-digit ppm HCPs, they are often labor-intensive and time-consuming with longer analysis time, making it not practical for routine analysis. The native digestion protocol introduced by Huang et al. [25] can remove the abundant antibodies by utilizing trypsin digestion under nondenaturing conditions. Under these conditions, the therapeutic protein derived from the CHO cell line is maintained largely intact while HCPs are digested. Thus, the dynamic range for HCP detection by MS is one to two orders of magnitude less than the traditional trypsin digestion sample preparation procedure. The native digestion protocol is a simple and faster approach and has been used in many published papers [18,22,26,27,28,29]. However, there is risk that the recovery of a small subset of low-level HCPs may not be favored during native digestion [30].

It is desirable to use a simple one-dimensional analytical flow LC-MS/MS method for HCP monitoring during the downstream purification processes of the biotherapeutic drugs and measure the HCP profile changes for routine analysis. In this study, we developed a highly sensitive, robust, and reproducible UHPLC-HRAM MS/MS method for HCP identification and monitoring using a Vanquish™ Flex UHPLC system(Thermo Fisher Scientific, Sunnyvale, CA, US) coupled with an Orbitrap™ Ascend BioPharma Tribrid™ mass spectrometer (Thermo Fisher Scientific, San Jose, CA, US). To address the dynamic issue, we used (1) native digestion adapted after Huang L et al. [25] to deplete biotherapeutics, (2) a long UHPLC C18 column (2.1 mm × 250 mm, 2.2 µm) for increasing mAb sample loading capacity to enhance method sensitivity while maintaining good separation efficiency, (3) an advanced Orbitrap mass analyzer for obtaining high sensitivity and fast scan speed, and (4) multiple database search algorithms (Sequest HT and CHIMERYS) included in the Proteome Discoverer™ 3.1 software for increasing the HCP identification coverage. The performance of the developed analytical flow UHPLC MS/MS method for HCP analysis was evaluated using the commercially available NIST monoclonal antibody (NISTmAb) reference material (RM 8671). In total, 234 HCPs were identified from NISTmAb with high confidence, which is comparable with the numbers of HCPs reported in the literatures for this sample using nanoflow separation [28,29]. Most of detected NISTmAb HCPs are in sub-ppm low concentration ranges [29], demonstrating that our method can be used to support the process development of therapeutic drugs through the HCP analysis of in-process samples. As a proof of concept, we applied the method to the in-house trastuzumab samples to assess HCP clearance efficiency while using a new POROS™ Caprylate Mixed-Mode Cation Exchange Chromatography resin (POROS Caprylate mixed-mode resin, Thermo Fisher Scientific, Bedford, MA, USA) for polishing purification. The identified number of HCPs was significantly decreased from 380 to 78 after the polish purification, and most of the remaining HCPs in the polish pool showed significantly decreased abundances, proving high HCP clearance efficiency of the POROS Caprylate mixed-mode resin.

## 2. Results

### 2.1. Development of a Sensitive and Robust UHPLC MS/MS Method for HCP Monitroing

#### 2.1.1. Workflow of the Developed UHPLC MS/MS Method

The commercially available NISTmAb (RM 8671) was used for method development and performance evaluation. The critical considerations during the method development are to make sure the method is robust, reproducible, easy to use, easy to maintain, and easily transferred from one production site to another for monitoring HCP clearance in the biotherapeutic drug production process. Figure 1 shows the method workflow. The major challenge for LC-MS-based methods is the wide dynamic range (five–six orders of magnitude) needed to detect HCPs at <10 ppm levels in the presence of the dominant therapeutic proteins. To address this challenge, Huang et al. introduced the native digestion protocol which maintains nearly intact antibody drugs while HCPs are digested [25]. By removing the minimally digested and intact antibody drugs from the sample, this sample preparation method effectively reduces the dynamic range to make it easier to detect the trace concentration of HCPs. In our method, we used this native mAb digestion protocol to decrease the intrasample dynamic range. Analytical flow rate (300 μL/min) was used for digest mixture separation. Since longer column length increases the resolving power for better peptide separation efficiency of the complex peptide mixtures and increases material loading capacity for increasing the absolute amount of HCP peptides loaded into the column, yielding high HCP peptide signals for MS detection, a 25 cm long UHPLC column was used to load a large amount of sample while maintaining separation efficiency to boost the method sensitivity with the analytical flow rate. In our experiments, 176 μg of the digested NISTmAb was loaded on the column and analyzed in triplicate. The total ion chromatograms of the triplicated LC-MS/MS runs are shown in Figure 2. Good separation efficiency was observed even with this large amount of mAb sample injection. The variation in retention time of eluted peaks was less than 0.2 min over three technical LC-MS runs. The data-dependent MS/MS (dd MS/MS) acquisition method using the Orbitrap detector was used to obtain clean and selective MS/MS data to ensure high-confidence HCP identification. In order to efficiently pick up low-abundance HCP peptides for MS/MS scans, each full MS scan (120k at 200 *m/z*) was followed by 15 MS/MS scans (30k at 200 *m/z*) of the most intense precursor ions with charge 2+–6+. To ensure that the quality of MS/MS data was maintained even at low concentration levels, a long maximum ion injection time (150 ms) was implemented for MS/MS scans to inject more precursor ions into the c-trap for HCD fragmentation. Proteome Discoverer 3.1 software was used for data processing using two search engines (Sequest HT and CHIMERYS™).

#### 2.1.2. Increasing HCP Identification Coverage by Using Multiple Database Research Algorithms

To evaluate if multiple search algorithms help to identify more HCPs, the data collected from the triplicate NISTmAb LC-MS runs were searched against a database containing all *Mus musculus* entries (17,728 entries, TaxID = 10090, 11 March 2024) extracted from UniProtKB/SwissProt using either a single search algorithm (Sequest HT/CHIMERYS) or two search algorithms (Sequest HT and CHIMERYS). We added the NISTmAb sequence to the database but filtered out the mAb peptides from the final results.

Acceptance criteria for positive HCP identifications included the following: (1) a representative unique protein in each protein group; (2) 1% false discovery rate (FDR) at the protein level; (3) at least two unique peptides at a confidence level of 95%. In total, 191 HCPs were identified by Sequest HT, and 217 HCPs were identified by CHIMERYS (Figure 3). The total number of HCPs identified from the NISTmAb increased to 234 by using both Sequest HT and CHIMERYS (Figure 3), demonstrating that more HCPs can be identified using multiple search algorithms. The full list of the identified HCPs using both Sequest HT and CHIMERYS can be found in the Appendix A.

### 2.2. Evaluating the Performance of the Developed UHPLC MS/MS Method for HCP Analysis

To evaluate the performance of our method for HCP analysis, we compared our NIST mAb HCP analysis results to a recently published report which used similar native digestion and 1D nanoflow separation for NISTmAb HCP identification and quantification [29]. We were able to identify 234 HCPs from the triplicate LC-MS/MS runs, which is more than the reported total of 133 HCPs identified by DDA and only slightly less than the reported total of 278 HCPs identified by DDA, FAIMS DDA, DIA, and GPF DIA. To estimate the method sensitivity to detect low-abundance HCPs, we also matched our identified HCPs to the reported quantification results (measured ng of HCP per mg of mAb, ppm) [29]. In total, 104 HCPs from our study have been matched to the reported HCPs with ppm values (Figure 4, Appendix A). Most matched HCPs had less than 1 ppm concentrations. The high sensitivity and high resolving power of the method enabled great quality of MS/MS data to be observed, even down to the 7 ppb low-concentration level (Figure 4), allowing confident HCP identification.

### 2.3. Applying the Developed UHPLC MS-MS/MS Method for HCP Profiling in Purification Process of Therapeutic Proteins

The results of the NISTmAb HCP analysis described above showcase the capability of our developed method to detect low abundance host cell proteins (HCPs) at sub-ppm levels. Our method offers extensive coverage in HCP identification, making it suitable for effective HCP identification and monitoring in therapeutic protein production processes. As a case study, we applied the method to an in-house-produced trastuzumab to assess the HCPs’ clearance efficiency of the newly released POROS Caprylate mixed-mode resin by tracking HCP profile changes from the samples collected before and after polish purification using this POROS Caprylate mixed-mode resin. Figure 5 shows the details of the two samples. Roughly 1 mg of each trastuzumab sample was used for trypsin digestion.

The digestion mixture of each sample was analyzed using the developed UHPLC MS/MS method in triplicate, respectively. A database containing all *Cricetulus Griseus* entries (89,053 entries, Tax ID = 10029, 15 March 2024) extracted from UniProtKB/TrEMBL was used for HCP identification. We added trastuzumab sequence in the database but filtered out the trastuzumab peptides from the final results. Figure 6 shows the total HCPs and unique peptides comparison between the two samples. In total, 380 HCPs including “high risk” HCPs such as Phospholipase B-like 2 (PLBL2) [6] were detected from the Protein A pool sample (Appendix A). The polish purification step removed most of the HCPs including PLBL2. Only 78 HCPs with much reduced abundance were identified from the polish pool sample (Appendix A).

High-risk HCPs are those that are immunogenic, biologically active, or enzymatically active with the potential to degrade either product molecules or excipients used in formulation, and some of them are difficult to remove [6]. Fifteen of the known “High Risk” HCPs were identified in the Protein A pool, and six of them, including PLBL-2, were removed with the polish purification (Figure 7). The previous study also identified 10 difficult-to-remove HCPs in CHO, which are difficult to remove during the downstream purification during co-elution or other protein–protein molecular interactions [7]. Seven of the known “difficult to remove HCPs” were identified in the Protein A pool, and five of them were removed with the polish purification (Figure 8).

For tracking the HCP abundance changes, the average of the top three peptide abundances (peak areas) were used without normalization. The data from the triplicate runs/sample were grouped and defined as the “control group” for the Protein A pool sample and the “sample group” for the polish pool sample. The abundance ratios of identified HCPs were calculated based on “grouped abundances of sample group/grouped abundances of control group”. The calculated abundance ratios of the individual HCPs between the Protein A pool sample group and the polish pool sample group are shown in the Appendix A.

In total, 80% of the identified HCPs showed ≤15% CVs (n = 3) in at least one of the sample groups. Figure 9 shows that the HCP abundance change trends for the representative “high risk” HCPs remained in the polish pool sample. Significant HCP abundance decreases (68–99.7% decrease rate) were observed for the remaining “high risk” HCPs. Regarding all HCPs, 37% of the identified HCPs showed a more than 90% abundance decrease rate, and 60% of the identified HCPs showed a more than 80% abundance decrease rate. Almost all remaining HCPs, except for the mitochondrial import inner membrane translocase subunit TIM50 and four keratins, showed abundance decrease trends (Appendix A), demonstrating that the POROS Caprylate mixed-mode resin was able to clear HCPs from the trastuzumab sample efficiently.

## 3. Discussion

We have developed a highly robust, highly sensitive, highly reproducible, easy-to-use, and easy-to-maintain analytical flow UHPLC HRAM MS/MS method for HCP analysis using an Orbitrap Ascend Biopharma Tribrid mass spectrometer coupled to a Vanquish Flex UHPLC system. Our method provided excellent peptide separation efficiency with high sample loading, resulting in sub-ppm-level HCPs being detected. By implementing the Orbitrap Ascend Biopharma Tribrid mass spectrometer as the detector, our method was able to pick up the majority of low-abundance HCP peptides for MS/MS data acquisition. By combining the database search results from both Sequest HT and CHIMERYS for HCP identification and relative quantification, we were able to identify 234 HCPs from a NISTmAb, which is more than the reported HCP numbers observed from the experiments using the data-dependent MS/MS approach with nanoflow rate separation [28,29]. Even with analytical flow rate separation, our method was very sensitive and able to detect low-abundance HCP peptides at levels as low as 0.007 ppm. This impressive sensitivity showcases the potential of our method for routine HCP profiling and monitoring during the downstream purification process of therapeutic protein drugs with sufficient HCP identification coverage and low detection limits of HCPs. In comparison to nano-LC-MS analysis, analytical flow LC-MS method has several advantages: (1) it is robust and reproducible; (2) it is easy to set up an HCP experiment using the standard LC and standard ESI source; (3) it is easy to maintain the instrument’s performance; (4) it is easy to transfer the instrument method without needing to learn how to use nano-LC-MS; and (5) there is no additional cost to purchase the nano-LC and nano-ESI source. Since most analytical groups supporting the mAb production process are already using analytical flow LC-MS techniques for their mAb characterization (such as peptide mapping and intact mass analysis), it should be easy for them to implement the method in their lab for HCP analysis.

While this method can be applied to any mAb samples in any process steps including drug substances and drug products for HCP analysis, it has some limitations. Since it uses a large number of therapeutic materials, it is not applicable to the type of therapeutics which do not have sufficient materials, such as AAV samples.

Plus, although the method transfer is easy, the method requires a high-performance HPLC analytical pump and an advanced Orbitrap mass spectrometer. Thus, it is not applicable for labs which do not have access to these instruments.

In our case study, we applied this method to identify and monitor HCPs from two trastuzumab samples (before and after the polish purification step) for assessing the HCP clearance efficiency of the polish purification material (POROS Caprylate mixed-mode resin). The method successfully identified 380 HCPs from the Protein A pool trastuzumab sample and 78 HCPs from the polish pool trastuzumab sample with high confidence and measured the HCP abundance changes between the two samples with good analytical precision. Based on the HCP analysis results: (1) 80% of individual HCPs identified in the Protein A pool samples was removed after the polish purification step; (2) among the 78 HCPs identified in the polish pool sample, 76 HCPs showed significant abundance decreases. Thus, we could conclude that the POROS Caprylate mixed-mode resin is highly efficient for clearing HCPs from trastuzumab samples.

The work described in this paper was focused on discovery HCP characterization and relative quantitation during the downstream mAb purification process. Although the developed analytical flow data-dependent MS/MS method can provide rapid HCP profiling across samples in different purification steps, it lacks absolute quantification information. As the next step of our study, we are planning to develop an isotopically labeled peptide kit which will include specific peptides from the known “High Risk” HCPs. The developed isotopic labeled peptide kit will be spiked into the digested samples collected in each purification step, and the absolute “High Risk HCP” amounts can be calculated by comparing the MS response from the native peptides and the known concentration of heavy-isotope-labeled HCP-specific peptides. We will also evaluate if the spiked-in isotopically labeled peptide kit can be used to estimate the absolute concentration ranges of other HCPs and further improve the quantitative precision.

## 4. Materials and Methods

### 4.1. Materials

The NISTmAb standard RM8671 (NIST, Gaithersburg, MD, USA) was purchased from Sigma-Aldrich (St. Louis, MO, USA). UltraPure™ 1 M Tris-HCI Buffer, Pierce™ Trypsin Protease MS grade, and Bond-Breaker TCEP Solution 0.5M solution were purchased from Thermo Fisher Scientific (Rockford, IL, USA). Two trastuzumab samples (one was from the pool of Protein A affinity chromatography at 10.81 mg/mL concentration level; another was from the pool of POROS Caprylate mixed-mode resin chromatography at 5.43 mg/mL concentration level) were generated internally.

### 4.2. Sample Preparation

All the mAb samples were digested with trypsin under nondenaturing conditions following the protocol published by Huang et al. [16]. Briefly, about 1 mg of each mAb sample was buffer exchanged to 50 mM Tris-HCl using 3k Amicon Ultra-0.5 Centrifugal Filter Unit (≈85 μL) and digested at 37 °C for two hours using a solution of trypsin enzyme at a 1:800 enzyme/protein ratio. The digest mixture was reduced with 2 μL of 0.5M TCEP for 10 min at 95 °C. The supernatant was acidified with 2 μL of 10%FA H_2_O and used for the HPLC MS/MS analysis.

### 4.3. HPLC MS/MS Analysis

All HCP samples were analyzed using an Orbitrap Ascend Biopharma Tribrid mass spectrometer (Thermo Scientific, San Jose, CA, USA) coupled to a Vanquish Flex UHPLC system (Thermo Scientific, Sunnyvale, CA, USA). Mobile phase A contained 0.1% FA in water, and mobile phase B contained 0.1% FA in acetonitrile. In total, 15 μL (≈176 μg of mAb if digestion was complete) of samples was loaded directly on an Acclaim™ VANQUISH™ C18 column, 2.1 × 250 mm, 2.2 μm (Thermo Scientific, Sunnyvale, CA, USA) and eluted at a flow rate of 300 µL/min following a linear gradient: 3% B at 0–1 min, 35% B at 90 min, and 85% B from 95 min to 100 min. The column wash cycle was implemented as follows: 3% B at 105 min, 85% B at 110 min, and 85% B from 110 min to 115 min. The column was re-equilibrated in 3% B from 115.1 min to 135 min. The column temperature was set to 60 °C. Each peak eluted from the column was analyzed using an electrospray ionization (ESI) source operating in positive peptide mode with spray voltage of 3.4 kV, sheath gas of 40, auxiliary gas of 10, capillary temperature of 320 °C, vaporizer temperature of 250 °C, and RF lens of 40%. Data were collected using data-dependent MS/MS experiment setups. For data acquisition, full MS scan data were acquired over a 350–1200 *m/z* range at a resolution of 120,000 (at 200 *m/z*), with a 300% normalized auto gain control (AGC) target value and a maximum injection time of 50 ms. Following each full MS scan, the top 15 most intense precursor ions with an intensity exceeding 5 × 10^3^ and charge state between two and six were automatically selected from the full MS spectrum with 2 amu isolation windows for higher-energy collisional dissociation (HCD) fragmentation at 28% normalized collision energy. MS/MS spectra were collected using Orbitrap detector with a fixed first mass of 120 *m/z* at a resolution of 30,000 (at 200 *m/z*), a 100% normalized AGC target value, and a maximum injection time of 150 ms.

### 4.4. Data Processing

Database searches for HCP identification and relative quantification were performed with Proteome Discoverer 3.1 using SEQUEST HT and CHIMERYS search algorithms against a database containing all *Mus musculus* entries (17,728 entries, TaxID = 10090, 11 March 2024) extracted from UniProtKB/SwissProt for NISTmAb sample and a database containing all *Cricetulus Griseus* entries (89,053 entries, TaxID = 10029, 15 March 2024) extracted from UniProtKB/TrEMBL for trastuzumab samples. For Sequest HT search setups, precursor ion mass tolerance was set to 10 ppm, and fragment ion mass tolerance was set to 0.02 Da. Trypsin was specified as the digestion enzyme during the database search with two missed cleavages allowed. Methionine oxidation and N-term modification of Met-loss and/or acetylation were set as variable modifications. The false discovery rate (FDR) was calculated using the Percolator node: 0.05 relaxed FDR and 0.01 strict FDR. For CHIMERYS search setups, Inferys_3.0.0_fragmentation prediction mode was used. Fragment ion mass tolerance was set to 20 ppm. Trypsin was specified as the digestion enzyme during the database search with two missed cleavages allowed. Cysteine carbamidomethylation was set as a static modification by default in this prediction mode.

For the HCP relative quantification, the average peak areas of the three most intense peptide abundances per HCP were used without normalization. The chromatographic alignment was set to 10 min and 10 ppm between the three technical replicates.

## Figures and Tables

**Figure 1 ijms-25-09687-f001:**
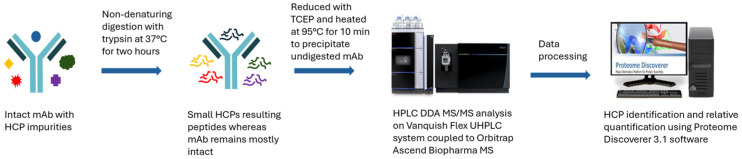
Workflow of the developed UHPLC MS/MS method for HCP analysis.

**Figure 2 ijms-25-09687-f002:**
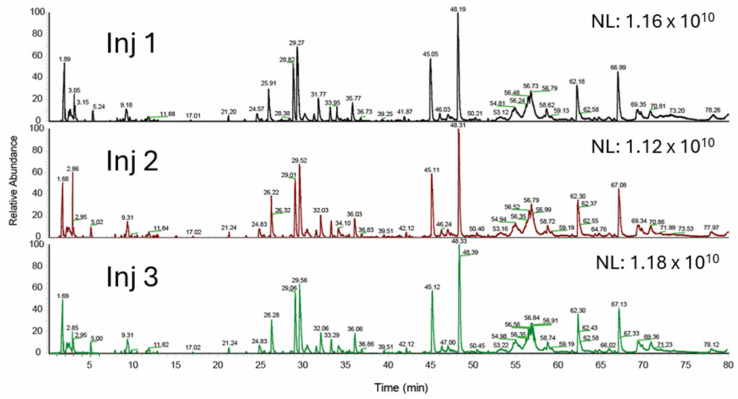
TIC of triplicate LC-MS runs of NISTmAb digest mixture.

**Figure 3 ijms-25-09687-f003:**
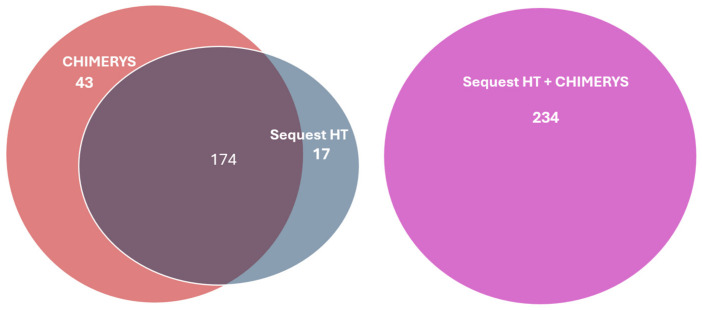
NISTmAb HCP id number comparison using one and two search algorithms.

**Figure 4 ijms-25-09687-f004:**
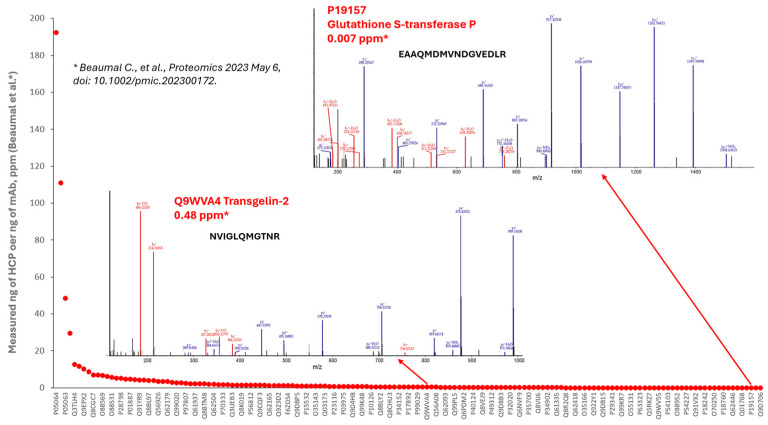
Identified HCPs with the concentration information shown in [29]. Most of them (>65%) were less than 1 ppm.

**Figure 5 ijms-25-09687-f005:**
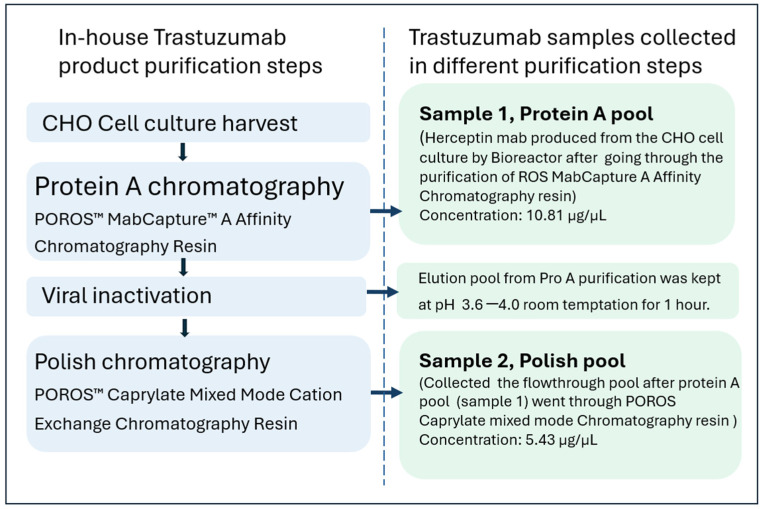
Information on the trastuzumab samples.

**Figure 6 ijms-25-09687-f006:**
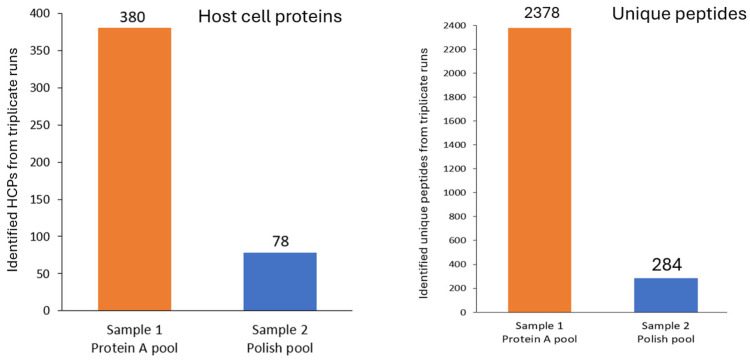
Comparison of the identified HCPs and unique peptides in the two samples collected before and after polish purification.

**Figure 7 ijms-25-09687-f007:**
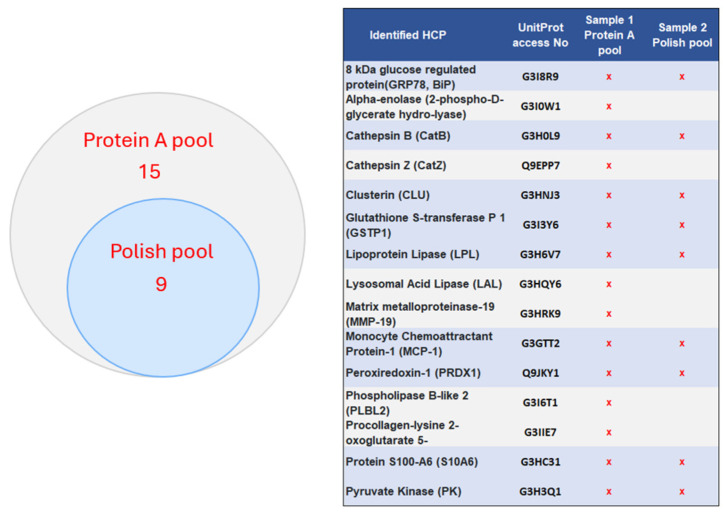
Identified high-risk HCPs in the two samples.

**Figure 8 ijms-25-09687-f008:**
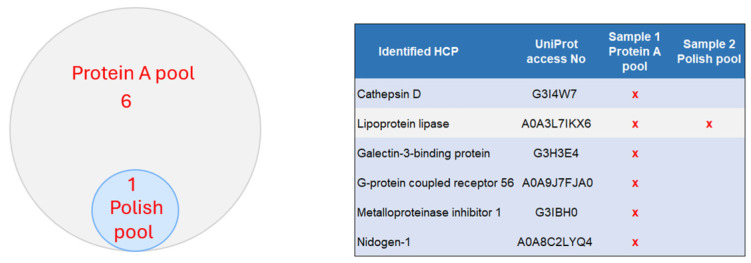
Identified difficult-to-remove HCPs in the two samples.

**Figure 9 ijms-25-09687-f009:**
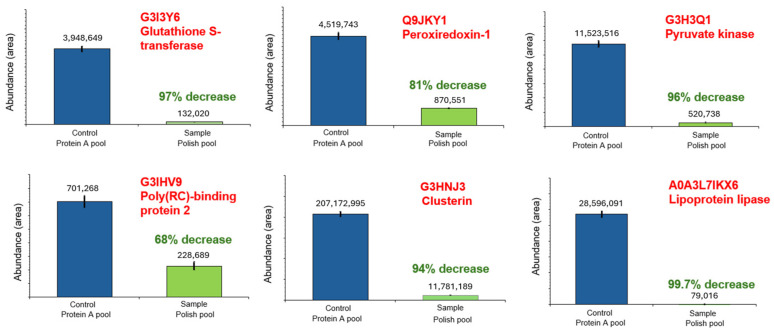
Representative high-risk HCP clearance trends.

## Data Availability

The data presented in this study are available on request from the corresponding author due to company policy.

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
