# Peer review of "Applying UHPLC-HRAM MS/MS Method to Assess Host Cell Protein Clearance during the Purification Process Development of Therapeutic mAbs"

_ijms, 2024, doi:10.3390/ijms25179687_

Round 1

Reviewer 1 Report

Comments and Suggestions for Authors

This article presents a significant advance in the analysis of HCPs using an optimised UHPLC-HRAM MS/MS method. The results are convincing and demonstrate the ability to detect proteins present at very low concentrations. The article discusses the development of a (LC-MSMS) method for the analysis of HCPs in biopharmaceutical production. The article highlights the major challenge of HCP analysis, namely the detection of low abundance proteins among dominant therapeutic proteins in order to maximise their identification. The authors have optimised sample prep, LC separation and data acquisition and processing. The evaluation of the developed method (NISTmAb standard) is a wise choice as it allows comparison with results published in the literature. The high confidence identification of HCPs (234 proteins) is an impressive result and demonstrates the ability of the pipeline to detect HCPs at low concentrations. Their pipeline is being applied to characterise HCPs in a commercial recombinant antibody. The method described in this article seems to be robust, highly sensitive, reproducible, easy to use and maintain. This makes it attractive for industrial use. However, a comparison with other existing methods for the analysis of HCPs in terms of sensitivity, reproducibility and ease of use would strengthen the position of this method. This would allow the developed method to be compared with those already established. To definitively establish the method, the transferability and potential limitations of the method would be necessary to further strengthen the validity and impact of this study.

Although not strictly necessary, these points could be improved.

Although several technical choices are mentioned, such as the use of a long C18 UHPLC column and native digestion, a more detailed explanation of why these specific choices were made would be beneficial. For example, why was a 250mm column chosen over a shorter one? Clearly, a more detailed explanation of why these specific choices were made would improve understanding. As the method uses the average abundance of the three most abundant peptides to track changes in HCP abundance, a discussion of potential normalisation techniques to improve the accuracy of relative abundance measurements would be beneficial.

The adaptation of the native digestion protocol of Huang et al. appears to be an effective strategy to reduce within-sample dynamics, but a more detailed explanation of why these specific choices were made would enhance understanding. In addition, a discussion of the potential limitations of this type of digestion, particularly in terms of reproducibility, inter-laboratory variability, would enhance understanding of the performance and applications of this method.

The pipeline appears robust but remains complex with several critical steps. A discussion on the ease of implementation of this method in an industrial environment would be useful.As  example,  any potential difficulties associated with the reproducibility or scaling-up of this method? Data showing the consistency of results between different batches, operators or production sites would reinforce the validity of the method. A more detailed comparison with other existing methods, including their advantages and disadvantages, could enhance the visibility of the proposed method. For example, how does this method compare in terms of cost, analysis time and ease of use?

Although the results are promising, a discussion of the potential limitations of the developed method would be useful. For example, is this method applicable to all types of biopharmaceuticals or are there exceptions? Or a discussion of the variability of the results obtained in terms of reproducibility between different batches of samples or between different operators would lend credibility to the claims of robustness and sensitivity and greatly enhance the article.

The use of Proteome Discoverer 3.1 software with two search engines (Sequest HT and CHIMERYS™) for data processing increases the identification coverage of HCPs, which is beneficial for comprehensive analysis. The combined use of the algorithms increased the total number of HCPs identified from 191 (Sequest HT) and 217 (CHIMERYS) to 234 when used together. This approach demonstrates that the use of multiple search algorithms can improve HCP identification coverage. However, a more detailed explanation of why these 2 algorithms were chosen over others available would be beneficial to understand the specific benefits of this combination. Was a comparison made, or could one be envisaged, with other common and more widely used search engines?

Author Response

Comment 1 from reviewer 1: Although several technical choices are mentioned, such as the use of a long C18 UHPLC column and native digestion, a more detailed explanation of why these specific choices were made would be beneficial. For example, why was a 250mm column chosen over a shorter one? Clearly, a more detailed explanation of why these specific choices were made would improve understanding. As the method uses the average abundance of the three most abundant peptides to track changes in HCP abundance, a discussion of potential normalization techniques to improve the accuracy of relative abundance measurements would be beneficial.

Response: . We have modified our manuscript to address the reviewer’s comments (page 3). In addition, we also added to the discussion that using a spiked-in isotopically labeled kit may help improve quantitation accuracy in the last of the discussion section (page 10).

Comment 2 from reviewer 1: The adaptation of the native digestion protocol of Huang et al. appears to be an effective strategy to reduce within-sample dynamics, but a more detailed explanation of why these specific choices were made would enhance understanding. In addition, a discussion of the potential limitations of this type of digestion, particularly in terms of reproducibility, inter-laboratory variability, would enhance understanding of the performance and applications of this method.

Response: We added to the manuscript the advantages of using native digestion based on the publication of Huange et al. in the method development section (page 3). We also added the potential limitation of the method in the introduction of the manuscript (page 2). Unfortunately, we did not have enough samples to evaluate the reproducibility of the native digestion method in this study.

Comment 3 from reviewer 1: The pipeline appears robust but remains complex with several critical steps. A discussion on the ease of implementation of this method in an industrial environment would be useful . As  example,  any potential difficulties associated with the reproducibility or scaling-up of this method? Data showing the consistency of results between different batches, operators or production sites would reinforce the validity of the method. A more detailed comparison with other existing methods, including their advantages and disadvantages, could enhance the visibility of the proposed method. For example, how does this method compare in terms of cost, analysis time and ease of use?

Response: Thanks for the comments. We added a discussion on the ease of implementation of this method in an industrial environment to the manuscript (page 10). We understood that there are so many different methods out there for HCP analysis, and it is a fit-in purpose choice to decide which method should be used per specific project. In addition, the scope of this paper is to develop an easy-to-use analytical flow method using widely accessible LCs. We added citations per approach in the introduction section.

Comment 4 from reviewer 1: Although the results are promising, . For example,  Or a discussion of the variability of the results obtained in terms of reproducibility between different batches of samples or between different operators would lend credibility to the claims of robustness and sensitivity and greatly enhance the article. For AAV samples, no enough materials.

Response: Thanks for the comments. We add to the manuscript some potential method limitations as suggested by the reviewer (page 11). We agree with the reviewer that AAV samples might not have enough materials, but it is not in the scope of this paper.

Comment 5 from reviewer 1: The use of Proteome Discoverer 3.1 software with two search engines (Sequest HT and CHIMERYS™) for data processing increases the identification coverage of HCPs, which is beneficial for comprehensive analysis. The combined use of the algorithms increased the total number of HCPs identified from 191 (Sequest HT) and 217 (CHIMERYS) to 234 when used together. This approach demonstrates that the use of multiple search algorithms can improve HCP identification coverage. However, a more detailed explanation of why these 2 algorithms were chosen over others available would be beneficial to understand the specific benefits of this combination. Was a comparison made, or could one be envisaged, with other common and more widely used search engines?

Response: We only have access to the Sequest HT and CHIMERYS search algorithms in Proteome Discoverer 3.1 software. Unfortunately, we cannot access other search engines to make a more comprehensive comparison. However, in the proteomics literature, there are multiple papers different search algorithms. Practical and Efficient Searching in Proteomics: A Cross Engine Comparison - PMC (nih.gov), Assessment and Comparison of Database Search Engines for Peptidomic Applications | Journal of Proteome Research (acs.org).

Reviewer 2 Report

Comments and Suggestions for Authors

Thé méthode described in this manuscript is of interest and worth publication in ijms. However the reviewer will suggest some modifications to improve the current form. In general all the caption of the figures should be completed so that the reader could understand without reference to the main text.

1) avoid abbreviations without defining them in the abstract (e.g. UHPLC-HRAM…

2) line 53: « Each approach has its advantages and disadvantages »  Could you please provide a two columns table describing for each approach the advantages and disadvantages.

3) lines 159-160: please simplify the sentence « Figure 6 shows the total HCPs and unique peptides identified in the two samples ». (the terme « identification number » is not very happy!).

4) line 165: delete « identification number »

5) figure 6: change the caption « Comparison of the identified HCPs and unique peptides in the two samples collected before and after polish purification »

7) figures 7 and 8: the left part is not necessary

caption of figure 7:  Identified high risk HCPs in the two samples

caption of figure 8: Identified difficult to remove HCPs in the two samples

8) figure 9: use « decrease » instead of « reduction »

9) lines 209-210: the abbreviations DS and DP are not useful since only cited once

10) Table 1 is not a table (!), only one column is present and the reviewer suppose that « Robuste, highly reproducible » is not the title of the column (why bold?) but one of the benefits of the method. It could be better to enumerate the benefits with boulet points. If the benefits derive from the method the authors developed, this should be stated in the title: « Benefits to use the analytical flow developed in this study for HCPs analysis »

11) to conclude the discussion please develop the last paragraph 

Author Response

Comment 1 from reviewer 2: avoid abbreviations without defining them in the abstract (e.g. UHPLC-HRAM…

Response: Corrected in the abstract section. (page 1)

Comment 2 from reviewer 2: line 53: « Each approach has its advantages and disadvantages »  .

Response: We have changed the text to reflect some of the advantages and disadvantages of each approach. We also added references for the readers who have an interest to learn more details about each approach in the introduction section (page 2-3).

Comment 3 from reviewer 2: lines 159-160: please simplify the sentence « Figure 6 shows the total HCPs and unique peptides identified in the two samples ». (the terme « identification number » is not very happy!).

Response: We made changes as the reviewer suggested.

Comment 4 from reviewer 2: line 165: delete « identification number »

Response: We made changes as the reviewer suggested.

Comment 5 from reviewer 2: figure 6: change the caption « Comparison of the identified HCPs and unique peptides in the two samples collected before and after polish purification »

Response: We made changes as the reviewer suggested.

Comment 6 from reviewer 2: figures 7 and 8: the left part is not necessary

Response: Thanks for the suggestion. Although it is a little redundant, we thought the diagrams gave the readers more visual effects and preferred to keep them.

Comment 7 from reviewer 2: caption of figure 7:  Identified high risk HCPs in the two samples

Response: We made changes as the reviewer suggested.

Comment 8 from reviewer 2: figure 9: use « decrease » instead of « reduction »

Response: We made changes as the reviewer suggested.

Comment 9 from reviewer 2: lines 209-210: the abbreviations DS and DP are not useful since only cited once

Response: As suggested, we deleted the abbreviations DS and DP (page 10).

Comment 10 from reviewer 2: Table 1 is not a table (!), only one column is present and the reviewer suppose that « Robuste, highly reproducible » is not the title of the column (why bold?) but one of the benefits of the method. It could be better to enumerate the benefits with boulet points. If the benefits derive from the method the authors developed, this should be stated in the title: « Benefits to use the analytical flow developed in this study for HCPs analysis »

Response: We deleted the table and added the benefits to the discussion section (page 10).

Comment 11 from reviewer 2: to conclude the discussion please develop the last paragraph

Response: Thanks for the suggestion. We have rewritten the last paragraph (page 10).

Reviewer 3 Report

Comments and Suggestions for Authors

The authors present a nice method on monitoring HCP clearance in biomanufacturing processes to generate mAbs. They offer a DDA-based, TopN, analytical-flow LC/MS method in contrast to nanoLC/MS methods observed in the literature.  This is not immediately obvious in the abstract or even in the introduction. 

"Polish Pool" is mentioned in the abstract but not explained.

What would be the rationale in going the analytical-flow route? A lot more sample would be required. 

Scientific names of organisms must be italicized.

On Pg-2, lines 48-51, the authors claim "The concentrations  of HCPs are significantly lower than the targeted therapeutic proteins, making it is very challenging to detect low abundant (<10 ppm) HCPs among the dominant therapeutic proteins." The authors need to provide information on why and how they think this intrasample dynamic range problem would be solved with an analytical-flow LC/MS method. What does "significantly lower" mean? For some analytes, ppb levels can pose a problem while for others, ppm is ok. How do the authors make that assessment for diverse HCPs?

The authors need to clarify what 'low abundance' means in the context of HCP clearance. How 'low' is 'low enough' for bioprocessing mAbs?

On Pg-2, lines 92-93, the authors state "In our experiments, roughly 176 μg of the digested NISTmAb was loaded on the column and analyzed in triplicate." That is a lot of protein sample. Would practitioners of such a protocol routinely have access to this amount of protein for analysis?

Why was DDA applied instead of DIA? How did authors ensure coverage of low-abundance HCPs given issues with DDA in this space?

Why was Top-N applied in place of TopSpeed?

The authors make mention of Ref # 32, which applied FAIMS in the analysis. Are the authors suggesting that FAIMS need not be used for this type of work?

On Pg-3 lines 100-102, the authors state "To ensure that the quality of MS/MS data is maintained even at low concentration levels, a long maximum ion injection time (150 ms) was implemented for MS/MS scans." The authors should clarify how this will help in the detection of low-abundance proteins given that DDA was used.

When SEQUEST HT and CHIMERYS were used together in data analysis, were they used in series or in tandem?

What is the cause of the differences observed in Figure 3?

Figure 5 is a bit confusing. Please restructure the figure to highlight every step of the procedure from start to finish.

Was mouse used as a host in method development and CHO as the host in the pilot experiment? Why did the authors not use the same host? It would have made for more robust deductions and conclusions.

The authors mentioned "UniProtKB/TrEMBL" as their source for databases. Where sub-taxonomies and protein variants considered? Did the authors include the sequence of the target mAb in the database? Why or why not?

The authors talk about 'reduced' abundance of HCPs in the polish purification step. They need to qualify and quantify what 'reduced' abundance means. What is considered "good enough" in the context of HCP clearance?

On Pg-5 lines 164-166, the authors state "The subset of 'high risk' HCPs [6] and 'difficult to remove HCPs' [7] identification number comparisons are shown in Figure 7 and 8, respectively." First of all, the sentence needs restructuring. Secondly, how does one define 'high risk' and 'difficult to remove'.

On Pg-6 lines 174-179, the authors state "For tracking the HCP abundance changes, the average of top 3 peptide abundances (peak areas) were used without normalization. The data from the triplicate runs/sample were grouped and defined as the 'control group' for the Protein A pool sample and the 'sample group' for the Polish pool sample. The abundance ratios of identified HCPs were calculated based on 'grouped abundances of sample group/grouped abundances of control group'." Please explain the rationale here. Did any HCPs 'increase' in abundance in the data? If so, can the authors comment on why?

The authors should explain what 'high risk' means in the context of HCP clearance (Pg-7, Fig-9).

Why do the authors believe their LC/MS method offered a deeper analysis of HCPs than nanoLCMS methods in the literature?

Pg-8, line-235. What is the benefit of applying non-denaturing conditions? Why is this important?

Did the authors consider 'trap-and-elute' in their LC method? Why or why not?

Comments on the Quality of English Language

English editing is required. Some sentences are verbose. Singular/Plural issues with some words. 

Author Response

Comment 1 from reviewer 3: The authors present a nice method on monitoring HCP clearance in biomanufacturing processes to generate mAbs. They offer a DDA-based, TopN, analytical-flow LC/MS method in contrast to nanoLC/MS methods observed in the literature. This is not immediately obvious in the abstract or even in the introduction.

Response: Thanks for the comments. We made changes in the abstract section to make it more straightforward (page 1).

Comment 2 from reviewer 3: What does “significantly lower” mean? For some analytes, ppb levels can pose a problem while for others, ppm is ok. How do the authors make that assessment for diverse HCPs? Define what the worlds mean.

The authors need to clarify what ‘low abundance’ means in the context of HCP clearance. How ‘low’ is ‘low enough’ for bioprocessing mAbs?

Response: “Significantly lower” in this paper means the HCP clearance rate was larger than 65%. Added more details about the HCP decrease in the manuscript (page 9). We did only relative quantification in this paper.  Without quantification results (ppm/ppb for individual HCPs),  how much “low” is  “low enough” discussion is out of the scope of this paper.

Comment 3 from reviewer 3: On Pg-2, lines 92-93, the authors state “In our experiments, μg of the digested NISTmAb was loaded on the column and analyzed in triplicate.” That is a lot of protein sample. Would practitioners of such a protocol routinely have access to this amount of protein for analysis?

Response: Generally, for biopharma applications like this one, there is enough amount of mAb material to run large loads (more than 100 g of mAbs material are generated for large-scale processing). So, in the large majority of cases, the sample amount is not an issue, and the method can be easily deployed.

Comment 4 from reviewer 3: Why was DDA applied instead of DIA? How did authors ensure coverage of low-abundance HCPs given issues with DDA in this space?

Response: We chose to use DDA because the MS/MS data are more selective and cleaner for highly confident HCP identification. We used NIST mAB to benchmark our HCP coverage, compared our HCP coverage with the published papers that used nano-flow for the NIST mAb analysis, and confirmed that we had comparable HCP ID coverage.

Comment 5 from reviewer 3: Why was Top-N applied in place of TopSpeed?

Response: We decided to run Top-N to have a higher number of MS1 for better quantification results.

Comment 6 from reviewer 3: On Pg-3 lines 100-102, the authors state “To ensure that the quality of MS/MS data is maintained even at low concentration levels, a long maximum ion injection time (150 ms) was implemented for MS/MS scans.” The authors should clarify how this will help in the detection of low-abundance proteins given that DDA was used. Explain how the longer injection time helps.

Response: Thanks for the suggestion. We added a more detailed explanation in the method development section (page 3).

Comment 7 from reviewer 3: When SEQUEST HT and CHIMERYS were used together in data analysis, were they used in series or in tandem?

Response: SEQUEST HT and CHIMERYS were used in parallel. The software got search results from each of them and later combined them.

Comment 8 from reviewer 3: What is the cause of the differences observed in Figure 3?

Response: SEQUEST HT and CHIMERYS have different search algorithms. Here are two references.  Practical and Efficient Searching in Proteomics: A Cross Engine Comparison - PMC (nih.gov), Assessment and Comparison of Database Search Engines for Peptidomic Applications | Journal of Proteome Research (acs.org).

Comment 9 from reviewer 3: Figure 5 is a bit confusing. Please restructure the figure to highlight every step of the procedure from start to finish.

Response: Thanks for the suggestion. We have made changes to Figure 5 to accommodate the suggestions.

Comment 10 from reviewer 3: Was mouse used as a host in method development and CHO as the host in the pilot experiment? Why did the authors not use the same host? It would have made for more robust deductions and conclusions.

Response: NIST mAb is a commercially available mAb standard and has been used for benchmarking HCP analysis in many published papers, which is why we used it for our method development. Unfortunately, we did not have any other mAb standard expressed in CHO cells available that could be used for benchmarking.

Comment 11 from reviewer 3: The authors mentioned “UniProtKB/TrEMBL” as their source for databases. Where sub-taxonomies and protein variants considered? Did the authors include the sequence of the target mAb in the database? Why or why not?

Response: We directly used extracted databases by using Tax ID 10029” for the CHO and “Tax ID 10090” for the mouse without considering sub-taxonomies and protein variants. We did include the target mAb in the database to check the sequence coverage of the target mAb, but we did not show the mAb coverage in the paper because we were focusing only on HCPs. We added this information to the manuscript. (Page 5, Page 7)

Comment 12 from reviewer 3: The authors talk about ‘reduced’ abundance of HCPs in the polish purification step. They need to qualify and quantify what ‘reduced’ abundance means. What is considered “good enough” in the context of HCP clearance?

Response: We calculated the percentage reduction for the peptides and added the information for the HCP abundance decrease in the result section (page 9). This work focuses on relative HCP clearance and “reduced abundance” meant the abundance of an individual HCP remained in the polish pool sample was lower than its abundance observed in the Protein A pool sample. We think that an HCP is efficiently cleared out if its abundance was decreased more than 50% through the polish purification step. However, the acceptable “good enough” in the context of HCP clearance is a hard discussion to answer without putting that into the context of a full risk assessment.

Comment 13 from reviewer 3: On Pg-6 lines 174-179, the authors state “For tracking the HCP abundance changes, the average of top 3 peptide abundances (peak areas) were used without normalization. The data from the triplicate runs/sample were grouped and defined as the ‘control group’ for the Protein A pool sample and the ‘sample group’ for the Polish pool sample. The abundance ratios of identified HCPs were calculated based on ‘grouped abundances of sample group/grouped abundances of control group’.” Please explain the rationale here. Did any HCPs’ increase’ in abundance in the data? If so, can the authors comment on why?

Response: We did not spike in any protein or peptide standards, which can be used for normalization in this work. For the quantification trends, all identified karatines showed slight increases in abundance, which may have been caused by sample preparation.

Comment 14 from reviewer 3: The authors should explain what ‘high risk’ means in the context of HCP clearance (Pg-7, Fig-9).

Response: Thanks for the comments. We added the explanations in the result section (page 7).

Comment 15 from reviewer 3: Why do the authors believe their LC/MS method offered a deeper analysis of HCPs than nanoLCMS methods in the literature?

Response: We optimized the HPLC and mass spectrometer conditions to allow high mAb material load and improve sensitivity. We also optimized the data processing process.

Comment 16 from reviewer 3: Pg-8, line-235. What is the benefit of applying non-denaturing conditions? Why is this important?

Response: We added detailed explanations in the method development section (page 3).

Comment 17 from reviewer 3: Did the authors consider ‘trap-and-elute’ in their LC method? Why or why not?

Response: No. We did not try. We did not have samples and instrument time for tryimg it out.

Reviewer 4 Report

Comments and Suggestions for Authors

The manuscript presented by Kiyonami and co workers shows the development of LC-MS/MÅš method for host cell proteins identification. The manuscript is well-prepared and well-written. 

However, in my opinion, additional explanation are needed before the paper is published. 

My questions are as follows:

- what is the novelty of the presented worki? Please define. 

- how was the LC separation method optimized?

- what was the collision energy in MS/MS analysis and how it was optimized to obtained the presented results?

The figures are of good quality. 

Test is well-organized. 

The cited literature is appropriate to to presented topic.

Comments on the Quality of English Language

Moderate English editing needed.

Author Response

Comment 1 from reviewer 4: what is the novelty of the presented work? Please define.

Response: This is a new developed analytical flow top 15 dd MS/MS method. It is the first method to use Orbitrap Ascend BioPharma Tribrid mass spectrometer for the HCP analysis and first method to apply multiple search engines for HCP identification. It is also the first paper to demonstrate the HCP clearance efficiency of the newly released POROS Caprylate mixed mode resin.

Comment 2 from reviewer 4: how was the LC separation method optimized?

Response: We did not do LC separation optimization. We used previously published works to define the LC conditions.

Comment 3 from reviewer 4: what was the collision energy in MS/MS analysis and how it was optimized to obtained the presented results?

Response: There is no need to do CE collision optimization for peptides on the Orbitrap Ascend MS. We just use the recommended 28% normalized collision energy.

Reviewer 5 Report

Comments and Suggestions for Authors

Dear authors,

Thank you for submitting your manuscript on "Applying UHPLC-HRAM MS/MS method to assess host cell protein clearance during the purification process development of therapeutic mAbs"

The manuscript is informative and well-written; however, I have some comments and questions.

1.      The manuscript's content and structure suggest that it could be more suitable for a technical note than a full article publication, which could be a great opportunity for a more focused and impactful publication.

2.      You have used the Protein A chromatography in the first step and as a comparison to the following chromatographic step with the mixed mode cationic separation. The benefit of the second separation step is shown in terms of cleaning the fraction originating from the Protein A step. However, I wonder if you have run an analysis including only the cation exchange step.

3.      Could you explain ”high risk” HCP?

4.      Please provide full access to the raw data by uploading to, e.g., PRIDE.

Kind regards.

Author Response

Comment 1 from reviewer 5: The manuscript’s content and structure suggest that it could be more suitable for a technical note than a full article publication, which could be a great opportunity for a more focused and impactful publication.

Response: We appreciate the suggestion. But we think that it is suitable for publication as a full article.

Comment 2 from reviewer 5: You have used the Protein A chromatography in the first step and as a comparison to the following chromatographic step with the mixed mode cationic separation. The benefit of the second separation step is shown in terms of cleaning the fraction originating from the Protein A step. However, I wonder if you have run an analysis including only the cation exchange step.

Response: The new resin was designed for polish purification purposes, and we did not run only the cation exchange step.

Comment 3 from reviewer 5: Could you explain ”high risk” HCP?

Response: Yes. Added explanation in the result section (page 7).

Comment 4 from reviewer 5: Please provide full access to the raw data by uploading to, e.g., PRIDE.

Response: Based on our company policy, We can provide full access to the raw data by request. Just need to have an email address.

Reviewer 6 Report

Comments and Suggestions for Authors

The study developed a UHPLC-HRAM MS/MS method using an Orbitrap Ascend BioPharma Tribrid mass spectrometer to identify and profile individual host cell proteins (HCPs) in biomanufacturing. Applied to Trastuzumab, this method evaluated the efficiency of the POROS™ Caprylate Mixed-Mode Cation Exchange Chromatography resin by comparing HCPs before and after purification. It successfully identified and quantified changes in individual HCPs between the Protein A pool and polish pool.

The authors mention that similar results were obtained without nanoLC, but show the actual comparative data.

You claim that the reproducibility is good, but three consecutive analyses do not show good reproducibility. Moreover, the retention times are off and the reproducibility is not good. Obtain validation data, including intra-day and inter-day reproducibility.

The authors state that there is an advantage of not using nanoLC, but the biggest disadvantage is that you have to use expensive equipment such as Orbitrap. A method should be developed and reported so that everyone can use it easily and conveniently.

Comments on the Quality of English Language

None.

Author Response

Comment 1 from reviewer 6: The authors mention that similar results were obtained without nanoLC, but show the actual comparative data.

Response: We used NISTmAb to benchmark our new method for HCP and compared our results with the published Nano LC NIST HCP analysis results, which also used native digestion protocol for sample preparation.

Comment 2 from reviewer 6: You claim that the reproducibility is good, but three consecutive analyses do not show good reproducibility. Moreover, the retention times are off and the reproducibility is not good. Obtain validation data, including intra-day and inter-day reproducibility.

Response: This paper focuses on the discovery of HCP and relative quantification. Validation, such as intra-day and inter-day reproducibility, is not part of the scope of this paper. For our triplicated runs, the retention time shifts of the eluted peaks were all less than 0.2 min. We consider it to be a good retention time reproducibility for this type of highly complex peptide mixture. We added this retention time shift window less than 0.2 min into the manuscript in the results section (page 4) for more specificity.

Comment 3 from reviewer 6: The authors state that there is an advantage of not using nanoLC, but the biggest disadvantage is that you have to use expensive equipment such as Orbitrap. A method should be developed and reported so that everyone can use it easily and conveniently.

Response: We just trying to develop a robust LC-MS method using our current instrument. We believe that our developed method is easy to use and easy to adapt for other users on other MS platforms.

Round 2

Reviewer 2 Report

Comments and Suggestions for Authors

The reviewer would like to thank the authors, they revised adequately the manuscript taking into account all the suggestions. The manuscript is really improved and can now be accepted for publication in ijms

Reviewer 6 Report

Comments and Suggestions for Authors

I have no more comments.